# Statistical Inference for Pairwise Graphical Models Using Score Matching

**Ming Yu**
mingyu@chicagobooth.edu

**Varun Gupta**
varun.gupta@chicagobooth.edu

**Mladen Kolar**[*]
mladen.kolar@chicagobooth.edu
University of Chicago Booth School of Business
Chicago, IL 60637

## Abstract

Probabilistic graphical models have been widely used to model complex systems and aid scientific discoveries. As a result, there is a large body of literature focused on consistent model selection. However, scientists are often interested in understanding uncertainty associated with the estimated parameters, which current literature has not addressed thoroughly. In this paper, we propose a novel estimator for edge parameters for pairwise graphical models based on Hyvärinen scoring rule. Hyvärinen scoring rule is especially useful in cases where the normalizing constant cannot be obtained efficiently in a closed form. We prove that the estimator is $\sqrt{n}$-consistent and asymptotically Normal. This result allows us to construct confidence intervals for edge parameters, as well as, hypothesis tests. We establish our results under conditions that are typically assumed in the literature for consistent estimation. However, we do not require that the estimator consistently recovers the graph structure. In particular, we prove that the asymptotic distribution of the estimator is robust to model selection mistakes and uniformly valid for a large number of data-generating processes. We illustrate validity of our estimator through extensive simulation studies.

## 1 Introduction

Undirected probabilistic graphical models are widely used to explore and represent dependencies between random variables. They have been used in areas ranging from computational biology to neuroscience and finance. See [7] for a recent review. An undirected probabilistic graphical model consists of an undirected graph $G = (V, E)$, where $V = \{1, \ldots, p\}$ is the vertex set and $E \subset V \times V$ is the edge set, and a random vector $X = (X_1, \ldots, X_p) \in \mathcal{X}^p \subseteq \mathbb{R}^P$. Each coordinate of the random vector $X$ is associated with a vertex in $V$ and the graph structure encodes the conditional independence assumptions underlying the distribution of $X$. In particular, $X_a$ and $X_b$ are conditionally independent given all the other variables if and only if $(a, b) \notin E$, that is, the nodes $a$ and $b$ are not adjacent in $G$. One of the fundamental problems in statistics is that of learning the structure of $G$ from i.i.d. samples from $X$ and quantifying uncertainty of the estimated structure.

---

[*]This work is supported by an IBM Corporation Faculty Research Fund at the University of Chicago Booth School of Business. This work was completed in part with resources provided by the University of Chicago Research Computing Center.

We consider a basic class of pairwise interaction graphical models with densities belonging to an exponential family $\mathcal{P} = \{p_\theta(x) \mid \theta \in \Theta\}$ with natural parameter space $\Theta$ and

$$\log p_\theta(x) = \sum_{a \in V} \sum_{k \in [K]} \theta_a^{(k)} t_a^{(k)}(x_a) + \sum_{(a,b) \in E} \sum_{l \in [L]} \theta_{ab}^{(l)} t_{ab}^{(l)}(x_a, x_b) - \Psi(\theta) + \sum_{a \in V} h_a(x_a), \quad x \in \mathcal{X} \subseteq \mathbb{R}^p.$$

(1)

The functions $t_a^{(k)}$, $t_{ab}^{(l)}$ are sufficient statistics and $\Psi(\theta)$ is the log-partition function. In this paper the support of the densities is either $\mathcal{X} = \mathbb{R}^P$ or $\mathcal{X} = \mathbb{R}_+^P$ and $\mathcal{P}$ is dominated by Lebesgue measure on $\mathbb{R}^p$. To simplify the notation, we will write $\log p_\theta(x) = \theta^\mathsf{T} t(x) - \Psi(\theta) + h(x)$ where $\theta \in \mathbb{R}^s$ and $t(x) : \mathbb{R}^p \mapsto \mathbb{R}^s$ with $s = \binom{p}{2} \cdot L + p \cdot K$. The natural parameter space has the form $\Theta = \{\theta \in \mathbb{R}^s \mid \Psi(x) = \log \int_{\mathcal{X}} \exp(\theta^\mathsf{T} t(x) dx) < \infty\}$. Under the model in (1), there is no edge between $a$ and $b$ in the corresponding conditional independence graph if and only if $\theta_{ab}^{(1)} = \cdots = \theta_{ab}^{(L)} = 0$. The model in (1) encompasses a large number of graphical models studied in the literature (see, for example, [7, 15] and referenced there in).

The main focus of the paper is on construction of an asymptotically normal estimator for parameters in (1) and performing (asymptotic) inference for them. We illustrate a procedure for construction of valid confidence intervals that have the nominal coverage and propose a statistical test for existence of edges in the graphical model with nominal size. Our inference results are robust to model selection mistakes, which commonly occur in ultra-high dimensional setting. Results in the paper complement existing literature, which is focused on consistent model selection and parameter recovery, as we review in the next section.

We use Hyvärinen scoring rule to estimate $\theta$, as in [15]. However, rather than focusing on consistent model selection we modify the regularized score matching procedure to construct a regular estimator that is robust to model selection mistakes and show how to use its asymptotic distribution for statistical inference. Compared to previous work on high-dimensional inference in graphical models [23, 2, 29, 11], this is the first work on inference in models where computing the normalizing constant is intractable.

**Related work.** Our work straddles two areas of statistical learning which have attracted significant research of late: model selection and estimation in high-dimensional graphical models, and high-dimensional inference. Our approach to inference for high-dimensional graphical models is based on regularized score matching. We briefly review the literature most relevant to our work, and refer the reader to a recent review article for a comprehensive overview [7].

**Graphical model selection:** Much of the research effort on graphical model selection has been done under the assumption that the data obeys the law $X \sim N(0, \Sigma)$ (Gaussian graphical models), in which case the edge set $E$ of the graph $G$ is encoded by the non-zero elements of the precision matrix $\Omega = \Sigma^{-1}$. More recently, [31] studied estimation of graphical models under the assumption that the node conditional distributions belong to an exponential family distribution (including, for example, Bernoulli, Gaussian, Poisson and exponential) via regularized likelihood (see also [13, 6, 30] and references therein). In our paper, we construct a novel $\sqrt{n}$-consistent estimator of a parameter corresponding to a particular edge in (1). As we mentioned earlier, this is the first procedure that can obtain a parametric rate of convergence for an edge parameter in a graphical model where computing the normalizing constant is intractable.

**High-dimensional inference:** Methods for construction of confidence intervals for low dimensional parameters in high-dimensional linear and generalized linear models, as well as hypothesis tests, have been developed in [32, 4, 28, 12]. These methods construct honest, uniformly valid confidence intervals and hypothesis test based on a first stage $\ell_1$ penalized estimator. [16, 23, 5] construct $\sqrt{n}$-consistent estimators for elements of the precision matrix $\Omega$ under a Gaussian assumption. We contribute to the literature on high dimensional inference by demonstrating how to construct estimators that are robust and uniformly valid under more general distributional assumptions than Gaussian.

**Score Matching estimators:** Score matching estimators were first proposed in [9, 10]. Score matching offers a computational advantage when the normalization constant is not available in closed-form making likelihood based approaches intractable. Despite its power, there have not been any results on inference in high-dimensional models using score matching. In [8], the authors use score matching for inference of Gaussian linear models (and hence for Gaussian graphical models) in low-dimensional setting. In [15], the authors use $\ell_1$ regularized score matching to develop consistent

estimators for graphical models in high-dimensional setting. We present the first high-dimensional inference results using score matching.

## 2 Score Matching

Let $X$ be a random variable with values in $\mathcal{X}$, and let $\mathcal{P}$ be a family of distributions over $\mathcal{X}$. A scoring rule $S(x, Q)$ is a real valued function that quantifies accuracy of $Q \in \mathcal{P}$ upon observing a realized value of $X$, $x \in \mathcal{X}$. There are a large number of scoring rules that correspond to different decision problems [20]. Given $n$ independent realizations of $X$, $\{x_i\}_{i \in [n]}$, one finds optimal score estimator $\widehat{Q} \in \mathcal{P}$ that minimizes the empirical score

$$\widehat{Q} = \arg \min_{Q \in \mathcal{P}} \mathbb{E}_n \left[ S(x_i, Q) \right]. \tag{2}$$

When $\mathcal{X} = \mathbb{R}^p$ and $\mathcal{P}$ consists of twice differentiable densities with respect to Lebesgue measure, the Hyvärinen scoring rule [9] is given as

$$S(x, Q) = (1/2)||\nabla \log q(x)||_2^2 + \Delta \log q(x) \tag{3}$$

where $q$ is the density of $Q$ with respect to Lebesgue measure on $\mathcal{X}$, $\nabla f(x) = \{\partial/(\partial x_j) f(x)\}_{j \in [p]}$ denotes the gradient, and $\Delta f(x) = \sum_{j \in [p]} \partial^2/(\partial x_j^2) f(x)$ the Laplacian operator on $\mathbb{R}^p$. This scoring rule is convenient for learning models that are specified in an unnormalized fashion or whose normalizing constant is difficult to compute. The score matching rule is proper, that is, $\mathbb{E}_{X \sim P} S(X, Q)$ is minimized over $\mathcal{P}$ at $Q = P$. Under suitable regularity conditions, the Fisher divergence between $P, Q \in \mathcal{P}$, $D(P, Q) = \int p(x)||\nabla \log q(x) - \nabla \log p(x)||_2^2 dx$, where $p$ is the density of $P$, is induced by the score matching rule [9]. For a parametric exponential family $\mathcal{P} = \{p_\theta \mid \theta \in \Theta\}$ with densities given in (1), minimizing (2) can be done in a closed form [9, 8]. An estimator $\widehat{\theta}$ obtained in this way can be shown to be asymptotically consistent [9], however, in general it will not be efficient [8].

Hyvärinen [10] proposed a generalization of the score matching approach to the case of non-negative data. When $\mathcal{X} = \mathbb{R}_+^p$ the scoring rule is given as

$$S_+(x, Q) = \sum_{a \in V} \left[ 2x_a \frac{\partial \log q(x)}{\partial x_a} + x_a^2 \frac{\partial^2 \log q(x)}{\partial x_a^2} + \frac{1}{2} x_a^2 \left( \frac{\partial \log q(x)}{\partial x_a} \right)^2 \right]. \tag{4}$$

For exponential families, the non-negative score matching loss again can be obtained in a closed form and the estimator is consistent and asymptotically normal under suitable conditions [10].

In the context of probabilistic graphical models, [8] studied score matching to learn Gaussian graphical models with symmetry constraints. [15] proposed a regularized score matching procedure to learn conditional independence graph in a high-dimensional setting by minimizing $\mathbb{E}_n \left[ \ell(x_i, \theta) \right] + \lambda ||\theta||_1$, where the loss $\ell(x_i, \theta)$ is either $S(x_i, Q_\theta)$ or $S_+(x_i, Q_\theta)$. For Gaussian models, $\ell_1$-norm regularized score matching is a simple but state-of-the-art method, which coincides with the method in [17]. Extending the work on estimation of infinite-dimensional exponential families [26], [27] study learning structure of nonparametric probabilistic graphical models using a score matching estimator. In the next section, we present a new estimator for components of $\theta$ in (1) that is consistent and asymptotically normal, building on [15] and [4].

## 3 Methodology

In this section, we propose a procedure that constructs a $\sqrt{n}$-consistent estimator of an element $\theta_{ab}$ of $\theta$. Our procedure is based on the three steps that we describe after introducing some additional notation. We start by describing the procedure for the case where $\mathcal{X} = \mathbb{R}^p$.

For fixed indices $a, b \in [p]$, let $q_\theta^{ab}(x) := q_\theta^{ab}(x_a, x_b \mid x_{-ab})$ be the conditional density of $(X_a, X_b)$ given $X_{-ab} = x_{-ab}$. In particular,

$$\log q_\theta^{ab}(x) = \langle \theta^{ab}, \varphi(x) \rangle - \Psi^{ab}(\theta, x_{-ab}) + h^{ab}(x)$$

where $\theta^{ab} \in \mathbb{R}^{s'}$ is a part of the vector $\theta$ corresponding to $\{\theta_a^{(k)}, \theta_b^{(k)}\}_{k \in [K]}$, $\{\theta_{ac}^{(l)}, \theta_{bc}^{(l)}\}_{l \in [L], c \in -ab}$ and $\varphi(x) = \varphi^{ab}(x) \in \mathbb{R}^{s'}$ is the corresponding vector of sufficient statistics with the dimension

$s' = 2K + 2(p-2)L$. Here $\Psi^{ab}(\theta, x_{-ab})$ is the log-partition function for the conditional distribution and $h^{ab}(x) = h_a(x_a) + h_b(x_b)$. Let $\nabla_{ab}f(x) = ((\partial/\partial x_a)f(x), (\partial/\partial x_b)f(x))^T \in \mathbb{R}^2$ be the gradient with respect to $x_a$ and $x_b$ and $\Delta_{ab}f(x) = ((\partial^2/\partial x_a^2) + (\partial^2/\partial x_b^2))f(x)$.

With this notation, we introduce the following scoring rule

$$S^{ab}(x, \theta) = (1/2)||\nabla_{ab} \log q_\theta^{ab}(x)||_2^2 + \Delta_{ab} \log q_\theta^{ab}(x) = (1/2)\theta^T \Gamma(x)\theta + \theta^T g(x), \quad (5)$$

where

$$\Gamma(x) = \varphi_1(x)\varphi_1(x)^T + \varphi_2(x)\varphi_2(x)^T \quad \text{and} \quad g(x) = \varphi_1(x)h_1^{ab}(x) + \varphi_2(x)h_2^{ab}(x) + \Delta_{ab}\varphi(x)$$

with $\varphi_1 = (\partial/\partial x_a)\varphi$, $\varphi_2 = (\partial/\partial x_b)\varphi$, $h_1^{ab} = (\partial/\partial x_a)h^{ab}$, and $h_2^{ab} = (\partial/\partial x_b)h^{ab}$. This scoring rule is related to the one in (3), however, rather than using the density $q_\theta$ in evaluating the parameter vector, we only consider the conditional density $q_\theta^{ab}$. We will use this conditional scoring rule to create an asymptotically normal estimator of an element $\theta_{ab}$. Our motivation for using this estimator comes from the fact that the parameter $\theta_{ab}$ can be identified from the conditional distribution of $(X_a, X_b) \mid X_{M_{ab}}$ where $M_{ab} := \{c \mid (a,c) \in E \text{ or } (b,c) \in E\}$ is the Markov blanket of $(X_a, X_b)$. Furthermore, the optimization problems arising in steps 1-3 below can be solved much more efficiently, as the problems are of much smaller dimension.

We are now ready to describe our procedure for estimating $\theta_{ab}$, which proceeds in three steps.

**Step 1:** We find a pilot estimator of $\theta^{ab}$ by solving the following program

$$\widehat{\theta}^{ab} = \arg\min_{\theta \in \mathbb{R}^{s'}} \mathbb{E}_n \left[ S^{ab}(x_i, \theta) \right] + \lambda_1 ||\theta||_1 \quad (6)$$

where $\lambda_1$ is a tuning parameter. Let $\widehat{M}_1 = M(\widehat{\theta}^{ab}) := \{(c,d) \mid \widehat{\theta}_{cd}^{ab} \neq 0\}$.

Since we are after an asymptotically normal estimator of $\theta_{ab}$, one may think that it is sufficient to find $\widetilde{\theta}^{ab} = \arg\min\{\mathbb{E}_n \left[ S^{ab}(x_i, \theta) \right] \mid M(\theta) \subseteq \widehat{M}_1\}$ and appeal to results of [21]. Unfortunately, this is not the case. Since $\widetilde{\theta}$ is obtained via a model selection procedure, it is irregular and its asymptotic distribution cannot be estimated [14, 22]. Therefore, we proceed to create a regular estimator of $\theta_{ab}$ in steps 2 and 3. The idea is to create an estimator $\widetilde{\theta}_{ab}$ that is insensitive to first order perturbations of other components of $\widetilde{\theta}^{ab}$, which we consider as nuisance components. The idea of creating an estimator that is robust to perturbations of nuisance have been recently used in [4], however, the approach goes back to the work of [19].

**Step 2:** Let $\widehat{\gamma}^{ab}$ be a minimizer of

$$(1/2)\mathbb{E}_n[(\varphi_{1,ab}(x_i) - \varphi_{1,-ab}(x_i)^T\gamma)^2 + (\varphi_{2,ab}(x_i) - \varphi_{2,-ab}(x_i)^T\gamma)^2] + \lambda_2 ||\gamma||_1. \quad (7)$$

The vector $(1, -\widehat{\gamma}^{ab,\mathrm{T}})^{\mathrm{T}}$ approximately computes a row of the inverse of the Hessian in (6).

**Step 3:** Let $\widetilde{M} = \{(a,b)\} \cup \widehat{M}_1 \cup M(\widehat{\gamma}^{ab})$. We obtain our estimator as a solution to the following program

$$\widetilde{\theta}^{ab} = \arg\min \mathbb{E}_n \left[ S^{ab}(x_i, \theta) \right] \quad \text{s.t.} \quad M(\theta) \subseteq \widetilde{M}. \quad (8)$$

Motivation for this procedure will be clear from the proof of Theorem 1 given in the next section.

**Extension to non-negative data.** For non-negative data, the procedure is slightly different. Instead of (5), as shown in [15], we instead define a different scoring rule $S_+^{ab}(x, \theta) = \frac{1}{2}\theta^T \Gamma_+(x)\theta + \theta^T g_+(x)$ with $\Gamma_+(x) = x_a^2 \cdot \varphi_1(x)\varphi_1(x)^T + x_b^2 \cdot \varphi_2(x)\varphi_2(x)^T$ and $g_+(x) = \varphi_1(x)h_1^{ab}(x) + \varphi_2(x)h_2^{ab}(x) + x_a^2\varphi_{11}(x) + x_b^2\varphi_{22}(x) + 2x_a\varphi_1(x) + 2x_b\varphi_2(x)$. Here $\varphi_{11} = (\partial^2/\partial x_a^2)\varphi$, and $\varphi_{22} = (\partial^2/\partial x_b^2)\varphi$. Now we can define $\widetilde{\varphi}_1 = x_a\varphi_1$ and $\widetilde{\varphi}_2 = x_b\varphi_2$. Then $\Gamma_+(x) = \widetilde{\varphi}_1(x)\widetilde{\varphi}_1(x)^T + \widetilde{\varphi}_2(x)\widetilde{\varphi}_2(x)^T$, which is of the same form as (5) with $\widetilde{\varphi}_1$ and $\widetilde{\varphi}_2$ replacing $\varphi_1$ and $\varphi_2$, respectively. Thus our three step procedure for non-negative data follows as before.

## 4  Asymptotic Normality of the Estimator

In this section, we outline main theoretical properties of our procedure. We start by providing high-level conditions that allow us to establish properties of each step in our procedure.

Assumption **M**. We are given $n$ i.i.d. samples $\{x_i\}_{i \in [n]}$ from $p_{\theta^*}$ of the form in (1). The parameter vector $\theta^*$ is sparse, with $|M(\theta^{ab,*})| \ll n$. Let

$$\gamma^{ab,*} = \arg\min \ \mathbb{E}[(\varphi_{1,ab}(x_i) - \varphi_{1,-ab}(x_i)^T \gamma)^2 + (\varphi_{2,ab}(x_i) - \varphi_{2,-ab}(x_i)^T \gamma)^2] \qquad (9)$$

and $\eta_{1i} = \varphi_{1,ab}(x_i) - \varphi_{1,-ab}(x_i)^T \gamma^{ab,*}$ and $\eta_{2i} = \varphi_{2,ab}(x_i) - \varphi_{2,-ab}(x_i)^T \gamma^{ab,*}$ for $i \in [n]$. The vector $\gamma^{ab,*}$ is sparse with $|M(\gamma^{ab,*})| \ll n$. Let $m = |M(\theta^{ab,*})| \vee |M(\gamma^{ab,*})|$.

The assumption **M** supposes that the parameter to be estimated is sparse, which makes estimation in high-dimensional setting feasible. An extension to approximately sparse parameter is possible, but technical. One of the benefits of using the conditional score to learn parameters of the model is that the sample size will only depend on the size of $M(\theta^{ab,*})$ and not on the sparsity of the whole vector $\theta^*$ as in [15]. The second part of the assumption states that the inverse of population Hessian is approximately sparse, which is a reasonable assumption since the Markov blanket of $(X_a, X_b)$ is small under the sparsity assumption on $\theta^{ab,*}$.

Our next condition assumes that the Hessian in (6) and (7) is well conditioned. Let $\phi_-(s, A) = \inf\{\delta^T A \delta / ||\delta||_2^2 \mid 1 \le ||\delta||_0 \le s\}$ and $\phi_+(s, A) = \sup\{\delta^T A \delta / ||\delta||_2^2 \mid 1 \le ||\delta||_0 \le s\}$ denote the minimal and maximal $s$-sparse eigenvalues of a semi-definite matrix $A$, respectively.

Assumption **SE**. The event

$$\mathcal{E}_{\mathrm{SE}} = \{\phi_{\min} \le \phi_-(m \cdot \log n, \mathbb{E}_n[\Gamma(x_i)]) \le \phi_+(m \cdot \log n, \mathbb{E}_n[\Gamma(x_i)]) \le \phi_{\max}\}$$

holds with probability $1 - \delta_{\mathrm{SE}}$ where $0 < \phi_{\min} \le \phi_{\max} < \infty$.

We choose to impose the sparse eigenvalue condition directly on $\mathbb{E}_n[\Gamma(x_i)]$ rather that on the population quantity $\mathbb{E}[\Gamma(x_i)]$. It is well known that the condition **SE** holds for a large number of models. See for example [24] and specifically [31] for exponential family graphical models.

Let $r_{j\theta} = ||\widehat{\theta}^{ab} - \theta^{ab,*}||_j$ and $r_{j\gamma} = ||\widehat{\gamma}^{ab} - \gamma^{ab,*}||_j$, for $j \in \{1, 2\}$, be the rates of estimation in steps 1 and 2. Under the assumption **SE**, on the event $\mathcal{E}_\theta = \{||\mathbb{E}_n[\Gamma(x_i)\theta + g(x_i)]||_\infty \le \lambda_1/2\}$ we have that $r_{1\theta} \le c_1 m\lambda/\phi_-$ and $r_{2\theta} \le c_2\sqrt{m}\lambda/\phi_-$. Similarly, on the event $\mathcal{E}_\gamma = \{||\mathbb{E}_n[\eta_{1i}\varphi_{1,-ab}(x_i) + \eta_{2i}\varphi_{2,-ab}(x_i)]||_\infty \le \lambda_2/2\}$ we have that $r_{1\gamma} \le c_1 m\lambda/\phi_{\min}$ and $r_{2\gamma} \le c_2\sqrt{m}\lambda/\phi_{\min}$ using results of [18]. Again, one needs to verify the two events hold with high-probability for the model at hand. However, this is a routine calculation under suitable tail assumptions. See for example Lemma 9 in [31].

The following result establishes a Bahadur representation for $\widetilde{\theta}_{ab}$.

**Theorem 1.** *Suppose that assumptions* **M** *and* **SE** *holds. Define $w^*$ with $w^*_{ab} = 1$ and $w^*_{-ab} = -\gamma^{ab,*}$, where $\gamma^{ab,*}$ is given in the assumption* **M**. *On the event $\mathcal{E}_\gamma \cap \mathcal{E}_\theta$, we have that*

$$\sqrt{n} \cdot \left(\widetilde{\theta}_{ab} - \theta^*_{ab}\right) = -\widehat{\sigma}_n^{-1} \cdot \sqrt{n}\mathbb{E}_n\left[w^{*,\mathrm{T}}\left(\Gamma(x_i)\theta^{ab,*} + g(x_i)\right)\right] + \mathcal{O}\left(\phi_{\max}^2\phi_{\min}^{-4} \cdot \sqrt{n}\lambda^2 m\right),$$
$$(10)$$

*where $\lambda = \lambda_1 \vee \lambda_2$ and $\sigma_n = \mathbb{E}_n[\eta_{1i}\varphi_{1,ab}(x_i) + \eta_{2i}\varphi_{2,ab}(x_i)]$.*

Theorem 1 is deterministic in nature. It establishes a representation that holds on the event $\mathcal{E}_\gamma \cap \mathcal{E}_\theta \cap \mathcal{E}_{\mathrm{SE}}$, which in many cases holds with overwhelming probability. We will show that under suitable conditions the first term converges to a normal distribution. The following is a regularity condition needed even in a low dimensional setting for asymptotic normality [8].

Assumption **R**. $\mathbb{E}_{q^{ab}}\left[||\Gamma(X_a, X_b, x_{-ab})\theta^{ab,*}||^2\right]$ and $\mathbb{E}_{q^{ab}}\left[||g(X_a, X_b, x_{-ab})||^2\right]$ are finite for all values of $x_{-ab}$ in the domain.

Theorem 1 and Lemma 9 together give the following corollary:

**Corollary 2.** *Suppose that the conditions of Theorem 1 hold. In addition, suppose the assumption* **R** *holds, $(m \log p)^2/n = o(1)$ and $\mathbb{P}(\mathcal{E}_\gamma \cap \mathcal{E}_\theta \cap \mathcal{E}_{\mathrm{SE}}) \to 1$. Then $\sqrt{n}(\widetilde{\theta}_{ab} - \theta^*_{ab}) \longrightarrow_D N(0, V) + o_p(1)$, where $V = (\mathbb{E}[\sigma_n])^{-2} \cdot \mathrm{Var}\left(w^{*,\mathrm{T}}\left(\Gamma(x_i)\theta^{ab} + g(x_i)\right)\right)$ and $\sigma_n$ is as in Theorem 1.*

We see that the variance $V$ depend on true $\theta^{ab}$ and $\gamma^{ab}$, which are unknown. In practice, we estimate $V$ using the following consistent estimator $\widehat{V}$,

$$e_{ab}^{\mathrm{T}}\left(\mathbb{E}_n[\Gamma(x_i)]_{\widetilde{M}}\right)^{-1}\left(\mathbb{E}_n\left[\left(\Gamma(x_i)\widetilde{\theta}^{ab} + g(x_i)\right)_{\widetilde{M}}\left(\Gamma(x_i)\widetilde{\theta}^{ab} + g(x_i)\right)_{\widetilde{M}}^{\mathrm{T}}\right]\right)\left(\mathbb{E}_n[\Gamma(x_i)]_{\widetilde{M}}\right)^{-1}e_{ab},$$

where $e_{ab}$ is a canonical vector with 1 in position of element $ab$. Using this estimate, we can construct a confidence interval with asymptotically nominal coverage. In particular,

$$\lim_{n\to\infty} \sup_{\theta^*\in\Theta} \mathbb{P}_{\theta^*}\left(\theta^*_{ab} \in \widetilde{\theta}_{ab} \pm z_{\alpha/2} \cdot \sqrt{\widehat{V}/n}\right) = \alpha + o(1).$$

In the next section, we outline the proof of Theorem 1. Proofs of other technical results are relegated to appendix.

## 4.1 Proof of Theorem 1

We first introduce some auxiliary estimates. Let $\widetilde{\gamma}^{ab}$ be a minimizer of the following constrained problem

$$\min \ \mathbb{E}_n\left[\left(\varphi_{1,ab}(x_i) - \varphi_{1,-ab}(x_i)^T\gamma\right)^2 + \left(\varphi_{2,ab}(x_i) - \varphi_{2,-ab}(x_i)^T\gamma\right)^2\right] \ \text{ s.t. } \ M(\gamma) \subseteq \widetilde{M} \tag{11}$$

where $\widetilde{M}$ is defined in step 3 of the procedure. Essentially, $\widetilde{\gamma}^{ab}$ is the refitted estimator from step 2 constrained to have the support on $\widetilde{M}$. Let $\widetilde{w} \in \mathbb{R}^{s'}$ with $\widetilde{w}_{ab} = 1$, $\widetilde{w}_{\widetilde{M}} = -\widetilde{\gamma}_{\widetilde{M}}$ and zero elsewhere. The solution $\widetilde{\theta}^{ab}$ satisfies the first order optimality condition $\left(\mathbb{E}_n\left[\Gamma(x_i)\right]\widetilde{\theta}^{ab} + \mathbb{E}_n[g(x_i)]\right)_{\widetilde{M}} = 0$. Multiplying by $\widetilde{w}$, it follows that

$$\widetilde{w}^{\mathrm{T}}\left(\mathbb{E}_n\left[\Gamma(x_i)\right]\widetilde{\theta}^{ab} + \mathbb{E}_n[g(x_i)]\right)$$
$$= (\widetilde{w} - w^*)^{\mathrm{T}}\mathbb{E}_n\left[\Gamma(x_i)\right]\left(\widetilde{\theta}^{ab} - \theta^{ab,*}\right) + (\widetilde{w} - w^*)^{\mathrm{T}}\left(\mathbb{E}_n\left[\Gamma(x_i)\theta^{ab,*} + g(x_i)\right]\right) +$$
$$w^{*,\mathrm{T}}\mathbb{E}_n\left[\Gamma(x_i)\right]\left(\widetilde{\theta}^{ab} - \theta^{ab,*}\right) + w^{*,\mathrm{T}}\left(\mathbb{E}_n\left[\Gamma(x_i)\theta^{ab,*} + g(x_i)\right]\right) \triangleq L_1 + L_2 + L_3 + L_4 = 0. \tag{12}$$

From Lemma 6 and Lemma 7, we have that $|L_1 + L_2| \leq C \cdot \phi_{\max}^2 \phi_{\min}^{-4} \cdot \lambda^2 m$. Using Lemma 8, the term $L_3$ can be written as $\mathbb{E}_n\left[\eta_{1i}\varphi_{1,ab}(x_i) + \eta_{2i}\varphi_{2,ab}(x_i)\right]\left(\widetilde{\theta}_{ab} - \theta^{ab,*}_{ab}\right) + \mathcal{O}\left(\phi_{\max}^{1/2}\phi_{\min}^{-2} \cdot \lambda^2 m\right)$. Putting all the pieces together completes the proof.

# 5 Synthetic Datasets

In this section we illustrate finite sample properties of our inference procedure on data simulated from three different Exponential family distributions. The first two examples involve Gaussian node-conditional distributions, for which we use regularized score matching. For the third setting where the node-conditional distributions follow an Exponential distribution, we use regularized non-negative score matching procedure. In each example, we report the mean coverage rate of 95% confidence intervals for several coefficients averaged over 500 independent simulation runs.

**Gaussian Graphical Model.** We first consider the simplest case of a Gaussian graphical model. The data is generated according to $X \sim N(0, \Sigma)$. We denote the precision matrix by $\Omega = \Sigma^{-1} = (w_{ab})$ (the inverse of covariance matrix).

For the experiment, we set diagonal entries of $\Omega$ as $w_{jj} = 1$, and we set the coefficients of the 4 nearest neighbor lattice graph according to $w_{j,j-1} = w_{j-1,j} = 0.5$ and $w_{j,j-2} = w_{j-2,j} = 0.3$. We set the sample size $n = 300$. Table 1 shows the empirical coverage rate for different choices of the number of nodes $p$ for four chosen coefficients. As is evident, our inference procedure performs remarkably well for the Gaussian graphical model studied.

**Normal Conditionals.** Our second synthetic dataset is sampled from the following exponential family distribution: $q(x|\boldsymbol{B}, \boldsymbol{b}, \boldsymbol{b}^{(2)}) \propto \exp\{\sum_{j\neq k}\beta_{jk}x_j^2 x_k^2 + \sum_{j=1}^p \beta_j^{(2)}x_j^2 + \sum_{j=1}^p \beta_j x_j\}$, where $\boldsymbol{b} = (\beta_1, \ldots, \beta_p)$ and $\boldsymbol{b}^{(2)} = (\beta_1^{(2)}, \ldots, \beta_P^{(2)})$ are $p$ dimensional vectors, and $\boldsymbol{B} = \{\beta_{jk}\}$ is a symmetric interaction matrix with diagonal entries set to 0. The above distribution is a special case of a class of exponential family distributions with normal conditionals, and densities that need not be unimodal [1]. This family is intriguing from the perspective of graphical modeling as, in contrast to the Gaussian case, conditional dependence may also express itself in the variances.

Table 1: Empirical Coverage for Gaussian Graphical Model

|         | $w_{1,2}$ | $w_{1,3}$ | $w_{1,4}$ | $w_{1,10}$ |
|---------|-----------|-----------|-----------|------------|
| $p = 50$  | 95.4% | 92.4% | 93.8% | 93.2% |
| $p = 200$ | 94.6% | 92.4% | 92.6% | 94.0% |
| $p = 400$ | 94.6% | 94.8% | 92.6% | 93.8% |

Table 2: Empirical Coverage for Normal Conditionals

|         | $\beta_{1,2}$ | $\beta_{1,3}$ | $\beta_{1,4}$ | $\beta_{1,10}$ |
|---------|---------------|---------------|---------------|----------------|
| $p = 100$ | 93.2% | 93.4% | 94.6% | 95.0% |
| $p = 300$ | 93.2% | 93.0% | 92.6% | 93.0% |

Table 3: Empirical Coverage for Exponential Graphical Model

|         | $\theta_{1,2}$ | $\theta_{1,3}$ | $\theta_{1,4}$ | $\theta_{1,10}$ |
|---------|----------------|----------------|----------------|-----------------|
| $p = 100$ | 92.0% | 90.0% | 90.0% | 92.4% |
| $p = 300$ | 92.6% | 92.0% | 92.2% | 92.4% |

For our experiment we set $\beta_j = 0.4$, $\beta_j^{(2)} = -2$, and we use a 4 nearest neighbor lattice dependence graph with interaction matrix: $\beta_{j,j-1} = \beta_{j-1,j} = -0.2$ and $\beta_{j,j-2} = \beta_{j-2,j} = -0.2$. Since the univariate marginal distributions are all Gaussian, we generate the data by Gibbs sampling. The first 500 samples were discarded as 'burn in' step, and of the remaining samples, we keep one in three. We set the number of samples $n = 500$. Table 2 shows the empirical coverage rate for $p = 100$ and $p = 300$ nodes. Again, we see that our inference algorithm behaves well on the above Normal Conditionals Model.

**Exponential Graphical Model.** Our final synthetic simulated example illustrates non-negative score matching for Exponential Graphical Model. Here the node-conditional distributions obey an exponential distribution, and therefore the variables take only non-negative values. Such exponential distributions are typically used for data describing inter-arrival times between events, among other applications. The density function is given by $q(x|\theta) \propto \exp\{-\sum_{j=1}^{p} \theta_j X_j - \sum_{j \neq k} \theta_{jk} X_j X_k\}$. To ensure that the distribution is valid and normalizable, we require $\theta_j > 0$, and $\theta_{jk} \geq 0$. Therefore, we can only model negative dependencies via the Exponential graphical model. For the experiment we choose $\theta_j = 2$, and a 2 nearest neighbor dependence graph with $\theta_{j,j-1} = \theta_{j-1,j} = 0.3$. We set $n = 1000$ and again use Gibbs sampling to generate data. The empirical coverage rate and histograms of estimates of four selected coefficients are presented in Table 3 and Figures 1 for $p = 100$ and $p = 300$, respectively.

We should point out that, in general, non-negative score matching is harder than regular score matching. For example, as shown in [15], to recover the structure from a regular Gaussian distribution

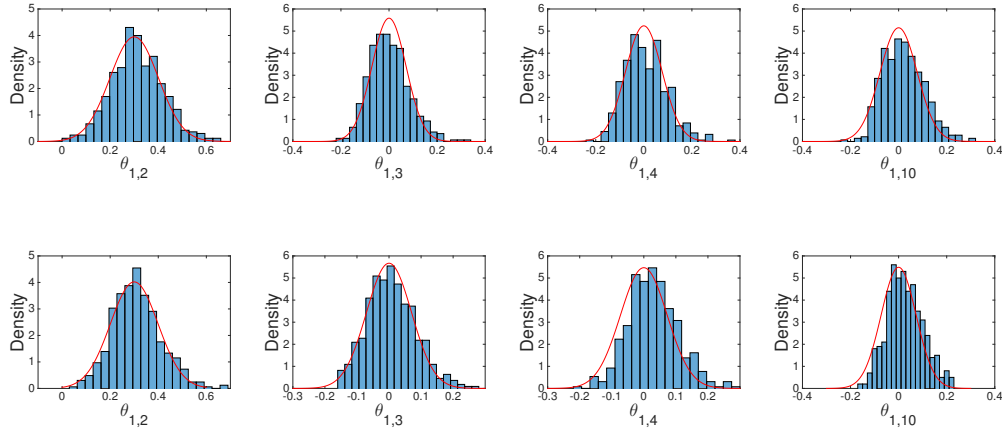

Figure 1: Histograms for $\theta$: the first row is for $p = 100$ and the second row is for $p = 300$

with high probability, a sample size about $\mathcal{O}(m^2 \log p)$ suffices, while to recover from non-negative Gaussian distribution, we need $\mathcal{O}(m^2 (\log p)^8)$, which is significantly larger. Therefore, we expect that confidence intervals for non-negative score matching would require more samples to give accurate inference. We can see this from Table 3, where the empirical coverage rate tends to be about 92%, rather than the designed 95% – still impressive for the not so large sample size. The histograms in Figures 1 show that the fitting is quite good, but to get a better estimation and hence better coverage, we would need more samples.

# 6   Protein Signaling Dataset

In this section we apply our algorithm to a protein signaling flow cytometry dataset. The dataset contains the presence of $p = 11$ proteins in $n = 7466$ cells. It was first analyzed using Bayesian Networks in [25] who fit a directed acyclic graph to the data, while [31] fit their proposed M-estimators for exponential and Gaussian graphical models to the data set.

Figure 2 shows the network structure after applying our method to the data using an Exponential Graphical Model. Since the data is non-negative and skewed, it can also be analyzed after log transformation as was done by [31] for fitting Gaussian graphical model. We instead learn the structure directly from the data without such a transformation. To infer the network structure, we calculate the $p$-value for each pair of nodes, and keep the edges with $p$-values smaller than 0.01. Estimated negative conditional dependencies are shown via red edges in the figure. Recall that the exponential graphical model restricts the edge weights to be non-negative, hence only negative dependencies can be estimated. From the figure we see that PKA is a major protein inhibitor in cell signaling networks. This result is consistent with the estimated graph structure in [31], as well as in the Bayesian network of [25]. In addition, we find significant dependency between PKC and PIP3.

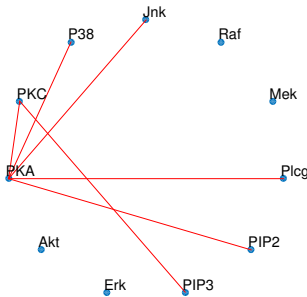

Figure 2: Estimated Structure of Protein Signaling Dataset

# 7   Conclusion

Driven by applications in Biology and Social Networks, there has been a surge in statistical learning models and methods for networks with large number of nodes. Graphical models provide a very flexible modeling framework for such networks, leading to much work in estimation and inference algorithms for Gaussian graphical models, and more generally for graphical models with node-conditional densities lying in Exponential family, in high dimensional setting. Most of this work is based on regularized likelihood loss minimization, which has the disadvantage of being computationally intractable when the normalization constant (partition function) of the conditional densities is not available in closed form. Score matching estimators provide a way around this issue, but so far there has been no work which provides inference guarantees for score matching based estimators for high-dimensional graphical models. In this paper we fill this gap for the case where score matching is used to estimate the parameter corresponding to a single edge at a time. An interesting future extension would be to perform inference on the entire model instead of one edge at a time as in the current paper. Another extension would be to extend our results to discrete valued data.

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
