[Reviews · NeurIPS 2016]

Reviewer 1

Summary

This paper develops an analysis in terms of asymptotic normality for a single edge parameter in graphical models using the Hyvarinen scoring rule. The Hyvarinen scoring rule means the log-partition function does not need to be known and consequently applies in settings where errors may be made in the model selection process. Section 4 provides the main theoretical result, asymptotic normality under two conditions, an assumption on the sparsity level and a sparse eigenvalue condition.

Qualitative Assessment

This paper addresses a relevant problem of proving asymptotic normality and consequently confidence intervals for learning edge parameters for pairwise graphical models. The score matching approach which has received recent interest overcomes the challenging of requiring knowledge of the log-partition function which is a useful contribution. The proofs appear to be correct but in Section 4.1, a slightly greater discussion on comparison of proof techniques in prior work that considers score-matching and confidence intervals in high-dimensional statistics would be useful. I believe this is a useful contribution to the graphical model learning and high-dimensional statistics communities.

Confidence in this Review

2-Confident (read it all; understood it all reasonably well)


Reviewer 2

Summary

This paper studies the problem of estimating parameters in a pairwise graphical model and constructing confidence intervals for the parameters. As part of this an asymptotically normal estimator is constructed. The key progress made in this paper is that inference (i.e., confidence intervals) is done in a setting where computation of basic quantities (e.g. partition function) is intractable. (In contrast, for Gaussian graphical model one can efficiently compute the normalizing constant.) Specifically, an estimator based on Hyvarinen score is given for estimation of a single edge in the pairwise undirected graphical model. The new scoring rule uses the conditional density of two variables given the rest. A first step forms a preliminary Markov blanket for a pair of variables, and the estimator then re-optimizes over parameter value, which has a sort of decoupling effect. The estimator is shown to satisfy asymptotic normality subject to regularity assumptions on the model (Corollary 2). Finally, experiments on simulated data show the estimator behaves as one would expect, and an experiment on protein signaling data agrees with results from previous papers analyzing that dataset. The estimator is

Qualitative Assessment

This paper was a pleasure to read: the results address a problem of significant interest, the writing is excellent, and the overall level of scholarship is high with context and background carefully placed within the broader literature. The proofs are fairly simple, making good use of existing technology developed in the cited work, but the estimator is novel. It would be nice to remark if possible on what does the assumption SE (line 176) really mean as far as modeling power of the model.

Confidence in this Review

2-Confident (read it all; understood it all reasonably well)


Reviewer 3

Summary

The paper presents a novel inference procedure for pairwise graphical models based on score matching. The proposed method is applicable to a variety of graphical models, going beyond the typical Gaussian case. Theory is very elegant. Illustrations using simulated data and real data are also very clear and convincing. Overall, this is an excellent paper with a novel method and good presentation.

Qualitative Assessment

A few minor concerns are: (1) In Corollary 2, one condition is that (s\log p)^2/n = o(1). Here is s the same as the one defined on page 2 line 33? If so, then s is a very large parameter. I wonder is the s in this Corollary be the sparsity parameter m. (2) The simulations reveal very good performance of the proposed method. Do you have any results of comparing the current method with existing methods, especially since there are available works under the Gaussian graphical models settings? Notations: (1) In Theorem 1 and Corollary 2, what is the vector w^*? Although it can be seen from the proof later what w is, it might be good to introduce the notation while presenting the main results. (2) In Corollary 2 line 195 definition of V: it should be E[\sigma_n]^{-2}. (3) The first line of equation (12) on page 6: tilde w rather than tilde w ^*. (3) Lemma 7 statement of the main equation: it should be tilde w - w^*. (4) Lemma 8 proofs line 378 first line of the equation: it should be w^*. Grammatical errors: (1) line 97 page 3: shown to be asymptotically ... (2) line 254 page 7: We should point out that ...

Confidence in this Review

2-Confident (read it all; understood it all reasonably well)


Reviewer 4

Summary

This paper works on the construction of estimates and inferences for pairwise graphical models with exponential family distributions. The paper employs the score matching trick and combines it with the post double selection idea for high dimensional regression to construct the estimates. This procedure provides a root-n consistent estimate of the pair-wise parameters as well as asymptotic normality. This involves establishing a Bahadur representation of the root-n scaled centered estimate of the parameter estimates. The authors also provide simulation studies to provide empirical validity of their methods.

Qualitative Assessment

The paper is well written and tackles an interesting problem. I liked the clever combination of post double selection methods and score matching to construct such estimate. I have the following specify comments: 1. I believe the authors use "m" to denote sparsity as shown in Assumption M but use "s" in Corollary 2 (and also later) in Page 7. "s" is used to denote total number of parameters in Page 2 which is O(p squared). 2. Could the authors highlight the sparsity requirements in terms of degree of the precision matrix etc? is "m" of the same order as degree? 3. Are the results in this paper also valid for non-negative data?

Confidence in this Review

2-Confident (read it all; understood it all reasonably well)


Reviewer 5

Summary

The paper proposes a parameter estimator for pairwise Markov networks with continuous random variables based on the Hyvärinen scoring rule. It proves that it is \sqrt{n} consistent and asymptotically normal, which enables the construction of confidence intervals for the parameters. Experiments on synthetic data confirm the estimator's properties. An experiment on a protein signaling dataset produces results consistent with estimates produced by other methods.

Qualitative Assessment

This paper addresses the hard problem of dealing with the partition function in undirected graphical models. Work on the Hyvärinen scoring rule is promising because it only requires knowing a density function up to a normalizing constant. The technical work all appears sound, although I am not 100% confident that I haven't missed something. However, I think the paper is only narrowly applicable. Is there another distribution beyond the exponential graphical model that this method can be used for? I also do not understand the focus on Gaussian graphical models in the experiments section. The related work discusses several other methods that are applicable to Gaussian graphical models, but it seems the only difference discussed (lines 62-64) is that this method is applicable to models for which the partition function is intractable. I'm unclear on whether the authors are including Gaussian graphical models in that category. (I wouldn't.)

Confidence in this Review

1-Less confident (might not have understood significant parts)


Reviewer 6

Summary

The paper suggests a novel method to learn pair-wise graphical model defined over continuous but not necessarily Gaussian variables. The method consists in suggesting a new score function based on Hyavarinen scoring rule. Importantly, the authors built a strong theoretical foundation for the method - proving/claiming that the method is $\sqrt{n}$ consistent and asymptotically normal (where $n\to\infty$ is the number of samples).

Qualitative Assessment

Few questions/comments: 1) I wonder if the authors have anything to say about how the number of samples sufficient for achieving a fixed accuracy scales with the system size? According to the experiments the scaling does not look exponential (difficult). Can the authors prove it, or at least argue? 2) I would also like to see discussion of how the approach explained in the paper is related to approaches seeking for objectives beyond these based on sufficient statistics, see e.g. https://arxiv.org/abs/1411.6156, https://arxiv.org/abs/1605.07252 and references therein. 3) Comment on the relation between the new score function and standard pseudo-log-likelihood score function would be useful. In this regards, experimental illustrations would certainly benefit from comparison with some other methods, e.g. based on the pseudo-log-likelihood and/or Gaussian fitting.

Confidence in this Review

2-Confident (read it all; understood it all reasonably well)